# Publication authorship: A new approach to the bibliometric study of scientific work and beyond

**Steffen Blaschke** *

Department of Management, Society, and Communication, Copenhagen Business School, Frederiksberg, Denmark

* sbl.msc@cbs.dk

**Data Availability Statement:** A step-by-step research protocol on how to obtain the raw data and accompanying source code for illustration and analyses is publicly available on GitHub at https://

## Abstract

Bibliometric studies offer numerous ways of analyzing scientific work. For example, co-citation and bibliographic coupling networks have been widely used since the 1960s to describe the segmentation of research and to look the development of the scientific frontier. In addition, co-authorship and collaboration networks have been employed for more than 30 years to explore the social dimension of scientific work. This paper introduces publication authorship as a complement to these established approaches. Three data sets of academic articles from accounting, astronomy, and gastroenterology are used to illustrate the benefits of publication authorship for bibliometric studies. In comparison to bibliographic coupling, publication authorship produces significantly better intra-cluster cosine similarities across all data sets, which in the end yields a more fine-grained picture of the research field in question. Beyond this finding, publication authorship lends itself to other types of documents such as corporate reports or meeting minutes to study organizations, movements, or any other concerted activity.

## Introduction

Bibliometric studies use publication data to describe the segmentation of research and to look at the development of the scientific frontier. The seminal works of the 1960s and 1970s [1–3] built networks of publications (vertices) connected by co-citation or bibliographic coupling (edges). Starting in the 1980s [4, 5], scholars turn to the social dimension of scientific research by looking at networks of authors (vertices) connected by author co-citation or co-authorship (edges).

Several refinements have been made to both publication and social networks in recent years. For example, co-citation proximity analysis [6] posits that citations appearing closer together (e. g., within a paragraph) in a publication are more similar than those further apart (e. g., one in the introduction and one in the discussion). This idea balances out the shortcoming that all citations contribute equally to the establishment of a relation between two (co-) cited publications. Another example is author bibliographic-coupling [7], which estimates a

github.com/blaschke/publication_authorship
(doi:10.5281/zenodo.6652069).

**Funding:** The author received no specific funding for this work.

**Competing interests:** The author has declared that no competing interests exist.

relation between authors based on the overlap of the bibliographies found in their complete oeuvres. This approach effectively expands the intellectual structure of scientific research from single publications to entire life works, which arguably paints a more realistic picture of the scientific frontier.

Bibliometric studies name either publications (e. g., academic articles, research grants, scientific patents) or individuals (i. e., authors) as the vertices of a network. The edges of the network, in turn, are either citations or authorship. The combination of vertices and edges then accounts for a number of different bibliometric networks, whether they are co-citation, bibliographic coupling, author co-citation, author bibliographic-coupling, or co-authorship networks. Notably missing from the combination of vertices and edges is the idea that publications may be connected by authorship. I consequently call this combination *publication authorship*. It clearly denominates vertices as publications and edges between them as the authorship of these.

In the following, I discuss the theoretical foundation of publication authorship including the most prevalent differences to traditional approaches in bibliometric studies. Empirical data from management, physics, and medicine then illustrates publication authorship opposite to bibliographic coupling. More specifically, I draw on academic articles published between 2010 and 2019 in the top-10 journals in accounting, astronomy, and gastroenterology. As a first step in the research, descriptive statistics for both publication authorship and bibliographic coupling provide an overview of development of the literature in these three academic areas. I then apply a standard clustering algorithm and test its goodness-of-fit using the cosine similarity between article abstracts. These empirical illustrations and respective statistical analysis show that publication authorship yields a significantly better segmentation of research than bibliographic coupling in all three academic areas, which consequently points out a more fine-grained picture of the scientific frontier. Finally, I point out similarities and differences in the findings for each one of the three academic areas, discuss co-word analysis and Latent Dirichlet Allocation as two alternative approaches, and conclude with implications for the theory and practice of bibliometric studies.

## Theoretical considerations

A brief introduction to co-citation, bibliographic coupling, author co-citation, author bibliographic coupling, and co-authorship sets the stage for an elaboration of publication authorship. Fig 1 provides an overview of these altogether six bibliographic networks. Publications appear as rectangles and authors as circles. Citations are directed either from publications (co-citation and author co-citation) or to publications (bibliographic coupling and author bibliographic-coupling), whereas authorship is undirected (i. e., individuals (co-)author publications and publications are (co-)authored by individuals).

### The intellectual structure of scientific work

Up until the mid-1960s, direct citation and keyword analysis are the dominant methods of inquiry into the structure of academic work and the development of the scientific frontier. The concepts of co-citation and bibliographic coupling are first and foremost critical responses to these methods used in early bibliometric studies.

Following in the footsteps of de Solla-Price [2], Small [3] introduces co-citation as a measure of scientific similarity in 1973. He argues that the frequency with which two publications are cited together by other publications (i. e., co-citation) is a better measure than direct citation, which is limited by the need of an explicit reference from one to another publication. Co-citation then identifies the intellectual connections between publications based on their citation patterns. It singles out seminal works in a given academic area using their citation count

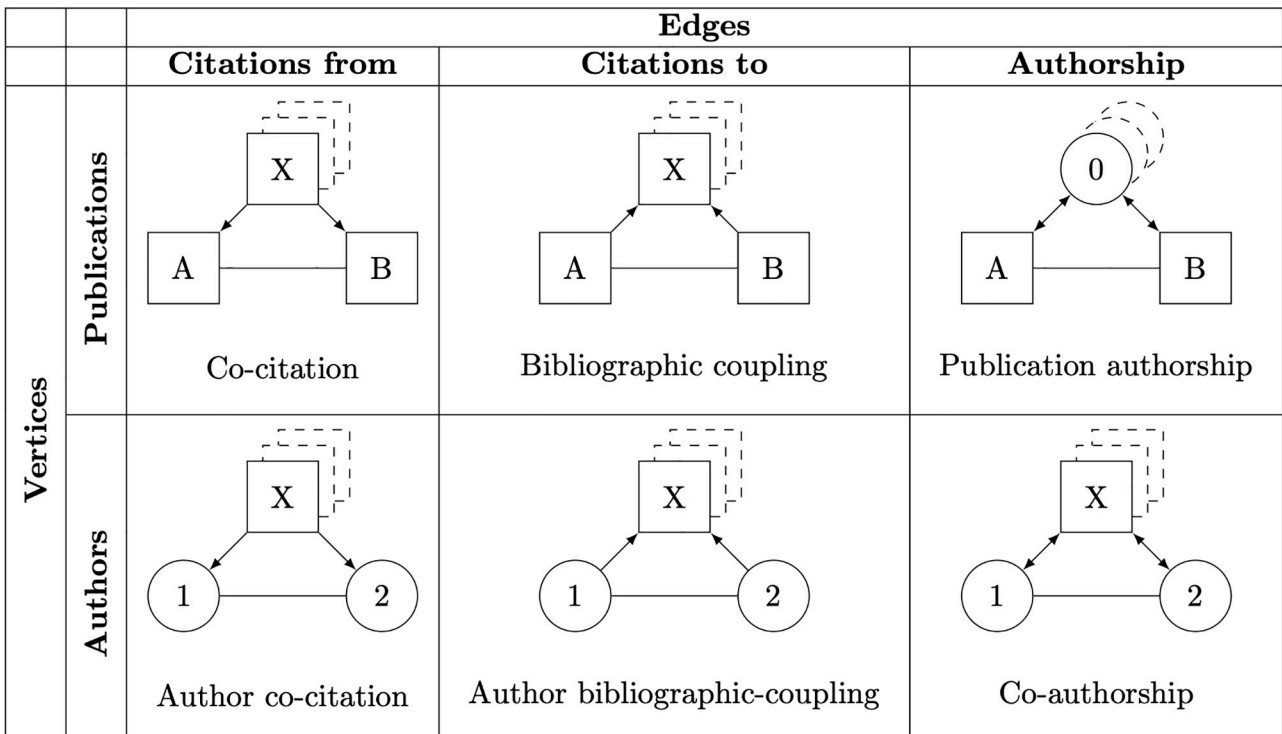

**Fig 1. Bibliometric networks.**

and tracks the development of intellectual ideas over time by looking at the evolution of clusters in co-citation networks [8].

Already ten years earlier, Kessler [1] introduces the concept of bibliographic coupling in 1963, which measures the similarity between two scientific publications based on the references they have in common. Bibliographic coupling effectively replaces earlier approaches (e. g., keyword analysis) to understand the development of academic areas. It is similar to co-citation in that it identifies structural properties of a given scientific field. However, where co-citation is more sensitive to the overall structure of a field, bibliographic coupling focuses on specific clusters of related publications. Co-citation maps the intellectual structure of an academic area and points to its research frontier, while bibliographic coupling relies on the similarity of publications to interpret core and peripheral works in a discipline.

Co-citation and bibliographic coupling both define publications as the vertices in a network. The edges in co-citation connect two publications (A and B) which are jointly cited by one or more other publications (X). They may be weighted by the number of publications which jointly cite the two. Conversely, bibliographic coupling connects two publications (A and B) which share common references to one or more other publications (X). The edges may be weighted by the the number of references two publications have in common. At the center of attention of both these bibliometric networks are the themes and topics of clusters of publications that make up schools of thought and push the scientific frontier.

Both co-citation and bibliographic coupling have been widely used in various fields of research such as biology, chemistry, physics, medicine, psychology, sociology, as well as computer, information, and management science. For example, Small et al. identify 71 emerging topics across all of science by combining direct citations and co-citations in publications from

2007 to 2010 [9]. They conclude that three non-exclusive forces drive research: scientific discovery, technological innovation, and exogenous events. On a side note, nearly all emerging topics contain highly cited papers, but only about 10 percent of highly cited papers are part of emerging topics. Jarneving complements bibliometric coupling with a complete-link cluster analysis [10] similar to previous work on co-citation clusters [11]. He tests this combination on a large multidisciplinary set of more than 600000 publications and 17 million references to estimate an optimal level of clustering that preservers core documents essential to the mapping of academic areas. His conclusion reveals but three large clusters of core documents. In a last example of research, Boyack and Klavans show which citation approach represents the intellectual structure of scientific work most accurately [12]. Their compelling comparison between (co-)citation and bibliographic coupling finds that the latter slightly outperforms the first approach with more coherent clusters to represent the scientific frontier.

## The social structure of scientific work

Beginning in the 1980s, bibliographic studies turn to the social dimension of scientific work. Author co-citation [4], author bibliographic-coupling [7], and co-authorship [5] are similar to co-citation and bibliographic coupling in that the main interest of any analysis is still the structure of scientific work. The key difference is that author co-citation, author bibliographic coupling, and co-authorship all focus on the social structure as opposed to the intellectual structure.

In author co-citation, two authors relate to each other if their works are frequently cited by other authors. In author bibliographic coupling, two authors relate to each other if they are frequently cited together in the same set of references. In addition, the concept of co-authorship allows for the study of collaborative relationships between authors in publications. Tracing these social structures provides insights into research communities and collaborations within and across scientific disciplines.

Author co-citation, author bibliographic-coupling, and co-authorship define authors as the vertices in a network. The edges in author co-citation connect two authors (1 and 2) who are jointly cited by one or more publications (X). They may be weighted by the number of publications which jointly cite the two. Conversely, the edges in author bibliographic-coupling connect two authors (1 and 2) who jointly cite one or more publications (X). The edges may be weighted by the number of publications two authors jointly cite. Finally, co-authorship connects two authors (1 and 2) who collaborate on one or more publications (X). The edges may be weighted by the number of publications two authors have in common. Clusters of authors stand in for schools of thought. Sometimes they are further grouped by affiliation or place to see which university or country is pushing the scientific frontier. Instead of a focus on the themes and topics of clusters of publications, the center of attention shifts to clusters of scientific collaboration among authors.

Similar to co-citation and bibliographic coupling, bibliometric studies of the social structure of scientific work span across various academic disciplines. For example, White and McCain study the social structure of information science [13]. They submit the top 120 authors most frequently cited in twelve key journals from 1972 through 1995 to author co-citation analysis. Their findings yield automatic classifications relevant to the history of the field including the most canonical authors. In a combination of co-authorship and bibliographic coupling, Biscaro and Giupponi examine citations counts of academic articles [14]. Their study based on 5585 publications from a variety of academic disciplines offers a number of findings, among which are: authors who collaborate with more authors tend to get more citations, and articles that use references from different strands of the literature tend to get more

citations. As a last example of research, Schubert and Glänzel take a look at country-by-country co-authorship to find that location, culture, and language determine clusters of mutually strong preferences in geopolitical areas such as Central Europe, Scandinavia, or the Far East [15]. The United States, unsurprisingly, enjoy universal co-authorship preference.

More comprehensive reviews of the theory and practice of bibliometric studies are found in Borgman and Furner [16], Mingers and Leydesdorff [17], and Donthu et al. [18].

## Combining the intellectual and social structure of scientific work

Publication authorship takes inspiration from the above discussed approaches to the analysis of scientific work. On the one hand, it defines publications as the vertices of a network similar to co-citation and bibliographic coupling. On the other hand, it takes authors as the basis of a definition of edges as authorship similar to co-authorship. The edges in publication authorship then connect two publications (A and B) which are authored by one or more individuals (0). They may be weighted by the number of authors two publications have in common. Publication authorship keeps the focus on the themes and topics of publications to describe the segmentation of research and the development of the scientific frontier. At the same time, it accounts for the social dimension of scientific work with clusters of publications emerging from the collaboration among authors.

Publication authorship may appear as simply another combination of vertices and edges that fills a void in the roster of approaches to the analysis of scientific work. However, it firmly rests with the theoretical argument of a communicative constitution of social systems [19]. The theory suggests that any form of documentation or record (e. g., academic publications, corporate reports, meetings minutes) is a condensate of the participation of individuals in communication [20]. In turn, individuals who participate in communication are common sources of information that connect communication event and episode [21]. Publication authorship follows exactly this line of argument. Scholars participate in academic discourse by authoring publications which, in turn, cluster to reflect the segmentation of research and the development of the scientific frontier [22].

Common to all the approaches in bibliometric studies is the idea that the relations among publications or authors present similarities in the underlying scientific work, which allows for the analysis of clusters of tightly coupled and central vertices (i. e., schools of thought and the scientific frontier). In particular, publication authorship assumes that two publications are similar to the extend that one or more scholars (co-)authors them. Since authors frequently specialize in a narrow field of research (e. g., behavioral economics or adolescent oncology), their publications are likely to present a narrow field of research, too (e. g., a behavioral economist is unlikely to work on transaction-costs issues and an adolescent oncologist rarely contributes to research on childhood obesity). Publication authorship is therefore more exclusive than co-citation and bibliographic coupling because the number of authors who collaborate on two publications is almost always smaller than then number of joint citations or common references. (None of the 27444 publications used in the empirical analysis of this paper had more authors than joint citations or common references.) At the same time, it is more inclusive than author co-citation and co-authorship because it includes both single-authored and co-authored publications.

Co-citation, bibliographic coupling, author co-citation, author bibliographic-coupling, co-authorship, and publication authorship all yield unique insights into scientific work. In the light of the similarities and differences among these and other approaches in bibliometric studies [23], publication authorship is closest to bibliographic coupling, not least because it defines vertices as publications and, therefore, focuses on the themes and topics of these. The following

empirical illustrations pit publication authorship against bibliographic coupling to highlight differences in the segmentation of research and a consequently more detailed scientific frontier.

## Data

Three data sets of academic articles in accounting, astronomy, and gastroenterology provide the empirical basis for the illustrations of publication authorship. The choice of academic disciplines is motivated by the idea to pick examples that are independent of each other, which is a safe assumption for scientific work in management, physics, and medicine. Indeed, there are no cross-references among the three data sets and each one exhibits its own unique features such as, for example, a smaller average number of authors for accounting than in astronomy or gastroenterology, a larger dispersion of the number of authors in astronomy than in gastroenterology, and a larger average number of references in accounting than in the other two disciplines (Table 1). These and other idiosyncrasies of each discipline reflect in the below analysis, of course. A smaller average number of authors on publications in accounting immediately translates to a lower density in respective bibliometric networks, and so on. The point of the empirical illustrations, however, is to compare bibliographic coupling to publication authorship across different academic disciplines, and not to compare disciplines to each other. Thus, I can safely report that data sets of academic articles in marketing, political science, and cancer research yield similar illustrations.

The data sets are compiled and downloaded from Elsevier's abstract and citation database Scopus. They comprise of academic articles published in the ten years between 2010 and 2019 in one of the top-10 journals for accounting, astronomy, and gastroenterology (see Table 17 in the S1 Appendix for an overview of journals). The journals are ranked according to their respective CiteScore in 2019. The data sets may be replicated following a step-by-step research protocol available on GitHub [24]. The R source code for the following illustrations of publication authorship can be found in the same location. Altogether, there are 5333 articles in accounting, 10817 articles in astronomy, and 11293 articles in gastroenterology.

As usual with publication data, the data sets require considerable cleaning before further analysis. This involves the removal of double entries (e. g., pre-prints), non-article publications (e. g., editorials, notes, letters, book reviews, errata), articles without an abstract or without references, and articles with anonymous authors. For later text mining, abstracts are stripped of punctuation, stop words, and numbers, multiple white-space characters are collapsed into one, and copyright notices are removed.

I compute networks for both bibliographic coupling and publication authorship in accounting, astronomy, and gastroenterology. Vertices represent academic articles. They are

**Table 1. Descriptive statistics.**

|  | Accounting | Astronomy | Gastroenterology |
|---|---:|---:|---:|
| Articles | 5333 | 10817 | 11293 |
| Authors | 6706 | 31568 | 58407 |
| Authors (mean) | 2.47 | 11.30 | 11.56 |
| Authors (s.d.) | 0.93 | 61.50 | 14.93 |
| References | 223639 | 367546 | 342230 |
| References (mean) | 58.03 | 50.39 | 40.82 |
| References (s.d.) | 30.83 | 46.13 | 26.79 |

**Table 2. Network measures.**

|  | Accounting | | Astronomy | | Gastroenterology | |
|---|---|---|---|---|---|---|
|  | **Bib. coup.** | **Pub. auth.** | **Bib. coup.** | **Pub. auth.** | **Bib. coup.** | **Pub. auth.** |
| Vertices | 5333 | 5333 | 10817 | 10817 | 11293 | 11293 |
| Isolates | 43 | 747 | 136 | 374 | 599 | 252 |
| Edges | 364151 | 16626 | 670640 | 232786 | 368923 | 328965 |
| Density | 0.03 | < 0.01 | 0.01 | < 0.01 | 0.01 | 0.01 |
| Transitivity | 0.30 | 0.58 | 0.34 | 0.23 | 0.34 | 0.35 |
| Assortativity | 0.32 | 0.57 | 0.33 | -0.02 | 0.33 | 0.21 |
| Clusters | 7 | 278 | 26 | 138 | 70 | 62 |

connected by edges either because they share one or more references in case of bibliographic coupling or because they have one or more authors in common in case of publication authorship. Therefore, the number of vertices is the same for both types of networks while the number of edges differs from one to the other (cf. Table 2). The difference in the number of edges among the networks already highlights the idiosyncrasies of each academic discipline. For example, the high average number of references in accounting leads to a more than 20 times higher edge count in bibliographic coupling than the low average number of authors in publication authorship. Conversely, the low average number of references in gastroenterology puts the number of edges for bibliographic coupling and publication authorship almost on par. Derivative measures such as network density (i. e., the ratio of the number of edges to the number of possible edges) differ accordingly.

Interestingly, network-level measures such as transitivity and assortativity do not follow the decreasing differences in the number of edges and network density from bibliographic coupling to publication authorship. Transitivity quantifies the probability that the adjacent vertices of a vertex are connected. In other words, it points out the probability that three articles form a triangle either because they share common references or authors. Transitivity reveals that the segmentation of research in bibliographic coupling is less dense than in publication authorship, it finds the inverse case to be true in astronomy, and shows a similar coefficient in gastroenterology. Assortativity quantifies the probability that a vertex connects to other vertices that are similar in one way or another. I use the degree of a vertex (i. e., the number of connections a vertex has to other vertices) to quantify the probability that an article with many common references or authors connects to other articles with many common references or authors. Assortativity shows an increase from bibliographic coupling to publication authorship in accounting and a decrease in astronomy and gastroenterology. These differences first and foremost highlight that academic areas are idiosyncratic in the way they conduct research. A low number of large research segments is most often associated with more loose connections among articles, whereas a high number of small research segments commonly calls for more dense connections among articles.

# Results

With a description of the data in place, I further investigate the differences between bibliographic coupling and publication authorship. I first compute clusters of articles, then estimate their goodness-of-fit to the data using a measure of cosine similarity, and finally discuss the segmentation of research and the development of the respective scientific frontier. These steps follow common practice in bibliometric studies (e. g., [25, 26]).

## Clustering

Transitivity and assortativity offer bird's-eye views of the clustering of networks. In order to compute clusters of vertices for bibliographic coupling and publication authorship across all three academic areas, I use a fast-greedy algorithm [27] widely employed in network analysis. The algorithm takes edge weights as an indicator of the strength of bibliographic coupling or publication authorship. I use the cosine similarity between a set of references or authors from publication *A* and a set of references or authors from publication *B*:

$$\text{cosine similarity} = \frac{|A \cap B|}{\sqrt{|A| \times |B|}} \tag{1}$$

The weight of the respective edge between two vertices is therefore the ratio of the number of references or authors the two publications *A* and *B* have in common, normalized by the square root of the product of the number of references or authors from the two publications *A* and *B*.

Clusters delimit subsets of articles that share similar theoretical insight or empirical evidence based on common references or common authors. They may be thought of as schools of thought or theoretical paradigms. Consider, for example, bibliographic coupling in accounting. Seven clusters describe the majority of research in the ten years from 2010 to 2019. Four of them share common topics such as banks, information, investors, liquidity, and stock. In contrast, cluster 4 leans towards references to entrepreneurship, innovation, and knowledge. To some extend, these topics adhere to different theoretical paradigms, ranging from economics to law and social science.

Bibliographic coupling is infused with a number of troubles that publication authorship hopes to remedy. Among these troubles is the misconception that common references provide a unanimous argument [28, 29]. While it is true that a majority of articles cites references to back up an argument, the same references may well be used to undermine it. Bibliographic coupling is therefore ill equipped to account for the quality of the argument by weighting common references.

Publication authorship addresses this shortcoming based on the notion that authors themselves stand in for a school of thought. Authors are more likely to work together because they complement each other in their theoretical ideas, methodological approaches, or empirical interests. Conversely, scholars of opposing schools of thought are unlikely to publish together. There are famous and rare exceptions to this, of course. For example, the academic debate between Habermas and Luhmann eventually led to a joint book publication that carefully elaborated on the commonalities and differences between Habermas' theory of communicative action and Luhmann's social systems theory [30]. However, most debates take place as an exchange of arguments in the form of alternating publications or lectures between scholars (e. g., Bohr and Einstein on quantum theory or Hawking and Penrose on time-reversal invariance). Bibliographic coupling draws these academic debates together because the respective articles share common references, whereas publication authorship separates the fields of research based on the authors' opposing schools of thought (i. e., disjoint authorship).

The number of clusters from bibliographic coupling to publication authorship jumps from seven to 278 clusters in accounting, still shows a steep increase from 26 to 138 clusters in astronomy, but slightly decreases from 70 to 62 clusters in gastroenterology (Table 2). In general, bibliographic coupling yields larger clusters that are more inclusive of opposing research, whereas publication authorship produces a more fine-grained picture of schools of thought, theoretical arguments, or fields of interest. Fig 2 shows the distribution of clusters for bibliographic coupling and publication authorship in accounting, astronomy, and gastroenterology. Opposite to the number of articles in each cluster (gray bars) stands the cumulative percentage

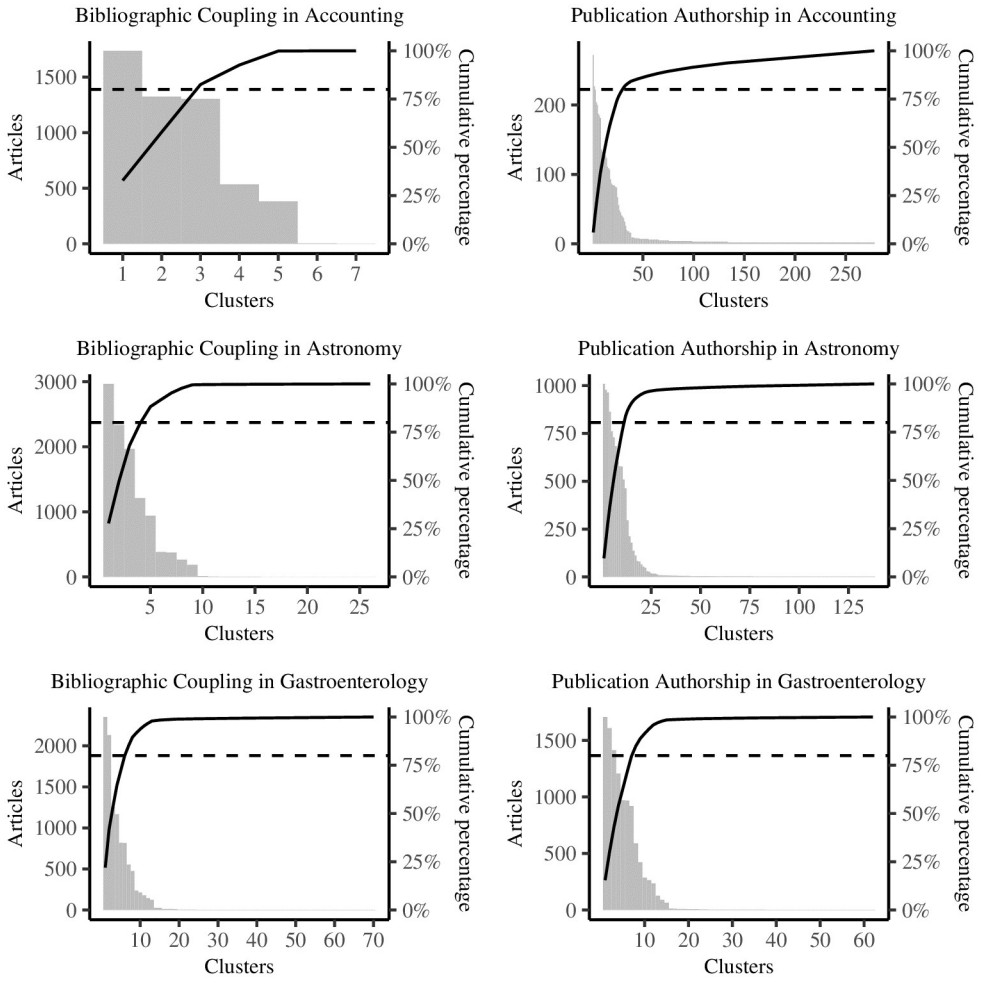

**Fig 2. Cluster distributions.**

of cluster sizes (solid black line) and the 80-percent cut-off (dashed black line). While this cut-off is arbitrary, it puts the focus on a limited number of clusters to tell a story about the segmentation of research and the development of the scientific frontier.

## Goodness-of-fit

Next, I look for evidence of how well clusters fit the bibliometric data. Given that two articles are assumed to be similar in their content based on common references or authors, I compute an additional similarity measure based on article abstracts. Following the above formula 1 for the cosine similarity between two attribute vectors of either references or authors, I compute the cosine similarity (i. e., edge weights) between attribute vectors of abstract terms of two articles (i. e., vertices). I then use the mean intra-cluster cosine similarity to compare the goodness-of-fit of clusters for bibliographic coupling and publication authorship in accounting, astronomy, and gastroenterology.

Fig 3 shows boxplots for the mean intra-cluster cosine similarities for bibliographic coupling and publication authorship in all three academic areas. In addition, I run a Mann-Whitney U test on the one-tailed alternative hypothesis that the means in publication authorship are greater than the means in bibliographic coupling. This alternative is true for all three

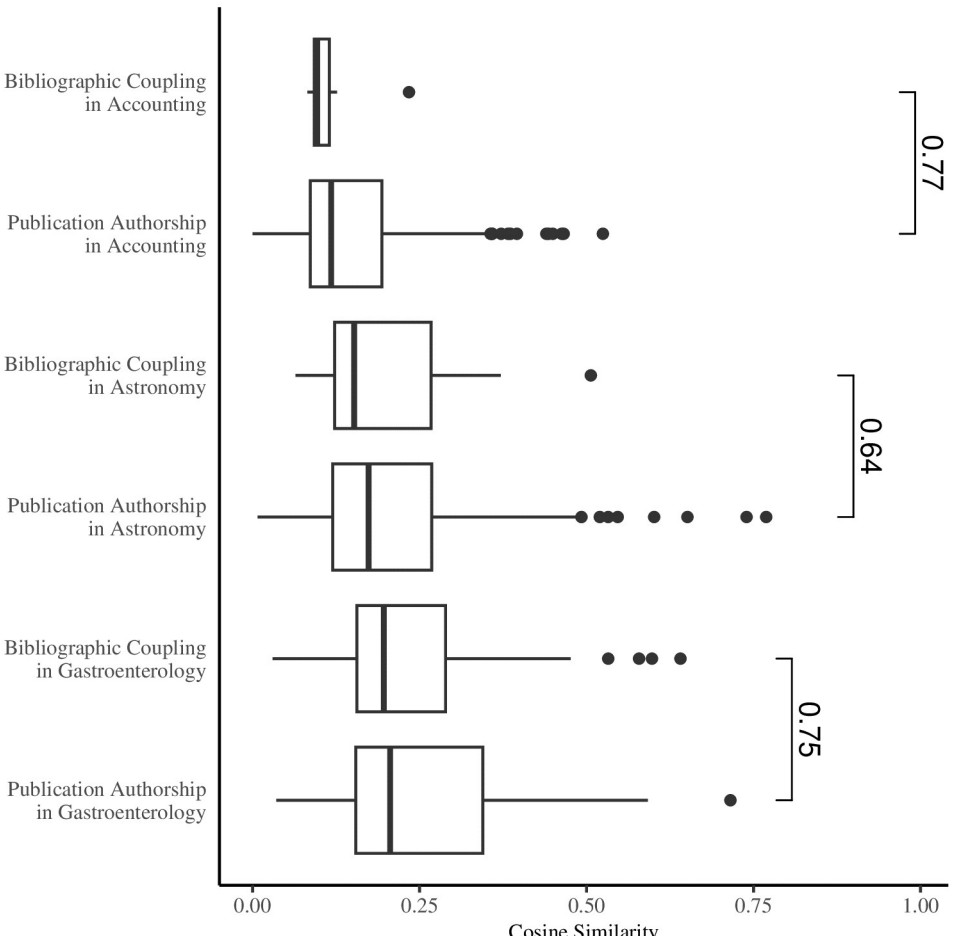

**Fig 3. Cluster boxplots.**

academic areas. Fig 3 additionally shows the corresponding non-parametric measure $p$, which can take on values between 0 or 1. The extreme values represent entirely separate distribution of means, whereas a $p$-value of 0.5 indicates a complete overlap. Accounting shows a difference in mean intra-cluster cosine similarities from bibliographic coupling to publication authorship at a $p$-value of 0.77. Although not as large a difference, publication authorship in astronomy also yields higher means at a $p$-value of 0.64. Finally, gastroenterology shows a difference between bibliographic coupling and publication authorship at a p-value of 0.75 despite a decrease in the number of clusters from one to the other. The results clearly show that the goodness-of-fit of clusters in publication authorship to the content of articles in questions is better than in bibliographic coupling.

## Research segmentation

I already established that bibliographic coupling is broader in the segmentation of research than publication authorship. The question now is, what additional insights does a more detailed picture yield? Again, I draw on networks to provide an answer for the segmentation of research and the development of the scientific frontier in accounting, astronomy, and gastro-enterology. The large numbers of articles and the bibliographic coupling or publication

authorship to connect them are prohibitive for any practical visualization. Therefore, I first collapse articles into clusters I already obtained with the help of the above presented algorithm. I then collapse bibliographic coupling or publication authorship between articles into respective relations between clusters and take the mean inter-cluster cosine similarity to weight these relations. Finally, I remove isolate clusters to focus the attention on the central component of each research field. Fig 4 shows six networks for bibliographic coupling and publication authorship in accounting, astronomy, and gastroenterology.

Bibliographic Coupling in Accounting

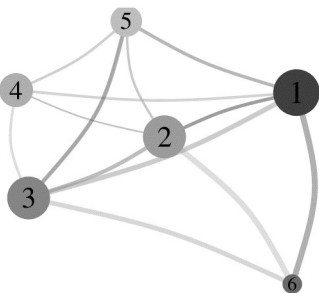

Publication Authorship in Accounting

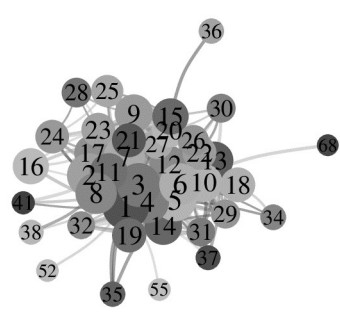

Bibliographic Coupling in Astronomy

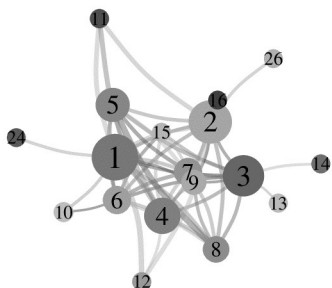

Publication Authorship in Astronomy

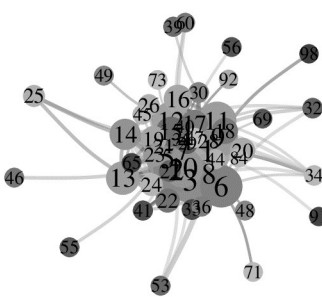

Bibliographic Coupling in Gastroenterology

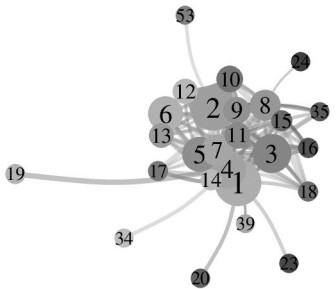

Publication Authorship in Gastroenterology

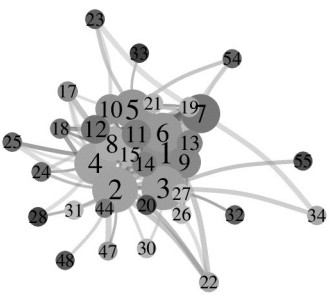

**Fig 4. Cluster networks.**

The size of the vertices indicates the (normalized) number of articles in each cluster, ranging from a minimum of two articles up to the biggest cluster with 2966 articles for bibliographic coupling in astronomy. The color of the vertices marks the mean age (in years) of articles in a cluster on a gray scale from the youngest cluster in light gray to the oldest cluster in dark gray. In like manner, the color and width of the edges indicates the mean cosine similarity between clusters on a gray scale from least similar relation in light gray to the most similar relation in dark gray. I use Kamada and Kawai's layout algorithm [31], which is among the most commonly used algorithms to position vertices and edges.

To describe the segmentation of research, I compute the term frequency-inverse document frequency (tf-idf) for article abstracts within each cluster for bibliographic coupling and publication authorship in accounting, astronomy, and gastroenterology to highlight the most prominent themes and topics. In addition to the visualization of the six networks (Fig 4), I report the number of articles, the mean and standard deviation of their age (in years), as well as the degree, betweenness, and closeness centrality for each cluster. A full glossary of respective technical terminology is found in the S2 Appendix.

Degree is the simplest measure of connectivity. It counts the number of edges a vertex has to other vertices. Betweenness and closeness centrality are frequently used measures in bibliographic studies where they signal interdisciplinarity and multidisciplinarity, respectively [25]. That is to say, the larger the number of shortest paths that go through a vertex (i. e., the more times a cluster sits in between others), the more that cluster may be considered to be interdisciplinary, and the smaller the average length of shortest paths from a vertex to all other vertices is (i. e., the closer a cluster is to others), the more that cluster may be considered to be multidisciplinary.

**Accounting.**   Bibliographic coupling in accounting comes about six connected clusters (Table 3). Already the three largest clusters (1, 2, and 3) combine more than 80 percent of all articles and broadly outline distinct research with only one shared tf-idf term (i. e., information; cf. Table 4). Cluster 5 also shares some common terms with the three largest clusters but is considerably smaller and younger, which may indicate a push of the scientific frontier. Cluster 4 sets itself apart with unique tf-idf terms such as research, universities, technology, innovation, and entrepreneurship. Nonetheless, bibliographic coupling paints a rather coarse picture for accounting.

Publication authorship, in turn, promises more details with the segmentation of research into 41 connected clusters. While there is considerable overlap in tf-idf terms among the top-ten clusters (e. g., cluster 2 shares seven terms with cluster 8 and five terms with clusters 3 and 7), some clusters exhibit exclusive terms that delineate unique lines of research (see Tables 5 and 6 for network measures and tf-idf terms). For example, cluster 2 centers on international financial reporting standards (ifrs), cluster 6 looks into high-frequency trading systems (hfts), and cluster 10 brings together venture capital (vc) and initial public offerings (ipo). Each of

**Table 3. Network measures for bibliographic coupling in accounting.**

| Cluster | Size | Age (mean) | Age (s.d.) | Degree | Betweenness | Closeness |
|--------:|-----:|-----------:|-----------:|-------:|------------:|----------:|
| 1 | 1737 | 6.08 | 2.74 | 5 | 0.67 | 0.20 |
| 2 | 1324 | 5.27 | 2.90 | 5 | 0.67 | 0.20 |
| 3 | 1304 | 5.67 | 2.87 | 5 | 0.67 | 0.20 |
| 4 | 535 | 4.63 | 2.74 | 4 | < 0.01 | 0.17 |
| 5 | 383 | 2.17 | 1.50 | 4 | < 0.01 | 0.17 |
| 6 | 5 | 5.80 | 3.11 | 3 | < 0.01 | 0.14 |

**Table 4. Top-ten tf-idf terms for bibliographic coupling in accounting.**

| Cluster | Top-ten tf-idf terms |
|---|---|
| 1 | banks, debt, information, ceo, capital, performance, accounting, corporate, stock, cash |
| 2 | accounting, performance, information, management, audit, paper, research, managers, control, based |
| 3 | returns, volatility, stocks, liquidity, stock, investors, risk, information, market, trading |
| 4 | research, universities, technology, knowledge, innovation, entrepreneurship, university, scientists, commercialization, paper |
| 5 | information, banks, returns, performance, liquidity, capital, debt, stock, effects, investors |
| 6 | tila, mentioning, overdraft, messages, takers, autodebit, couple, math, savvy, shrouding |

**Table 5. Network measures for publication authorship in accounting (10 largest clusters).**

| Cluster | Size | Age (mean) | Age (s.d.) | Degree | Betweenness | Closeness |
|---|---|---|---|---|---|---|
| 1 | 272 | 5.95 | 2.93 | 31 | 108.59 | 0.02 |
| 2 | 227 | 5.42 | 2.93 | 24 | 7.39 | 0.02 |
| 3 | 222 | 5.63 | 2.97 | 32 | 84.61 | 0.02 |
| 4 | 204 | 5.62 | 2.83 | 26 | 29.69 | 0.02 |
| 5 | 200 | 5.00 | 2.89 | 25 | 21.82 | 0.02 |
| 6 | 187 | 5.25 | 3.02 | 29 | 75.62 | 0.02 |
| 7 | 185 | 5.25 | 2.79 | 27 | 24.93 | 0.02 |
| 8 | 181 | 5.55 | 2.87 | 22 | 17.01 | 0.02 |
| 9 | 136 | 5.43 | 2.79 | 21 | 17.54 | 0.02 |
| 10 | 133 | 5.02 | 3.10 | 25 | 12.35 | 0.02 |

these three clusters marks a differentiation of research in accounting and thus a push of the scientific boundary.

**Astronomy.** Bibliographic coupling in astronomy shows a segmentation of research which is largely made up of four clusters (1, 2, 3, 4). These four clusters are closely connected to each other (Table 7) at the center of the network. They share tf-idf terms that any layperson would guess are descriptive of research in astronomy (e. g., galaxy, mass, star; Table 8). With an average overlap 7.5 tf-idf terms among them (most notably, clusters 2 and 3 share all top-ten terms, albeit in different order), the four largest clusters are too generic to constitute particular fields of interests in astronomy.

**Table 6. Top-ten tf-idf terms for publication authorship in accounting (10 largest clusters).**

| Cluster | Top-ten tf-idf terms |
|---|---|
| 1 | firms, audit, banks, funds, stock, equity, ceo, foreign, returns, capital |
| 2 | ifrs, firms, earnings, information, disclosure, stock, tax, financial, mandatory, returns |
| 3 | earnings, firms, investment, returns, equity, audit, stock, risk, investors, tax |
| 4 | firms, risk, investors, debt, returns, model, asset, leverage, cash, equity |
| 5 | risk, returns, volatility, model, consumption, firms, credit, asset, bidders, cash |
| 6 | trading, stocks, hfts, returns, liquidity, prices, stock, dark, price, dealers |
| 7 | tax, earnings, firms, audit, disclosure, corporate, investment, quality, credit, information |
| 8 | tax, earnings, firms, information, returns, investors, firm, analysts, disclosure, stock |
| 9 | auditors, audit, managers, performance, team, accounting, quality, auditor, effort, judgments |
| 10 | credit, vc, banks, bank, firms, ipo, ratings, liquidity, lenders, loan |

**Table 7. Cluster network measures for bibliographic coupling in astronomy (10 largest clusters).**

| Cluster | Size | Age (mean) | Age (s.d.) | Degree | Betweenness | Closeness |
|---|---|---|---|---|---|---|
| 1 | 2966 | 5.85 | 3.10 | 13 | 31.51 | 0.05 |
| 2 | 2334 | 4.02 | 2.58 | 13 | 38.01 | 0.05 |
| 3 | 1967 | 7.05 | 1.19 | 11 | 31.68 | 0.04 |
| 4 | 1211 | 5.79 | 3.06 | 11 | 7.01 | 0.04 |
| 5 | 942 | 5.66 | 2.95 | 8 | 0.14 | 0.04 |
| 6 | 382 | 4.77 | 2.92 | 9 | 5.83 | 0.04 |
| 7 | 375 | 5.15 | 2.86 | 9 | 0.68 | 0.04 |
| 8 | 265 | 5.44 | 3.14 | 8 | 0.14 | 0.04 |
| 9 | 186 | 3.16 | 2.16 | 7 | < 0.01 | 0.04 |
| 10 | 13 | 1.00 | 0.00 | 2 | < 0.01 | 0.03 |

Some smaller clusters are more unique in their contributions to the research field. For example, cluster 5 exhibits a large body of research on solar flares and cluster 9 features numerous studies on the formation of stars and other stellar objects. In the end, bibliographic coupling makes astronomy appear as if it was a field of research where perhaps only some newer or renewed interests (e. g., the smaller and younger cluster 9 opposite the older and larger cluster 4) are bound to push the scientific boundary.

Publication authorship splits research in astronomy into 56 connected clusters. The ten largest clusters make up almost 80 percent of all publications. This more fine-grained picture immediately reflects in the 49 unique top-10 tf-idf terms that describe the clusters, whereas bibliographic coupling only shows 34 unique terms (Tables 9 and 10).

A combination of tf-idf terms such as black, hole, and kev (kiloelectron volts) in cluster 9 then points to the latest research findings based on data from NASA's Nuclear Spectroscopic Telescope Array. In contrast, bibliographic coupling buries this research mainly in its largest cluster 1. A similar observation can be made for research on the formation of galaxies found in cluster 3. Next to the generic tf-idf terms such as galaxy, mass, and star, the additional term redshift specifically contributes to our understanding of an ever expanding universe where light from distant stellar objects shifts towards longer wavelength and, therefore, moves into the red end of the electromagnetic spectrum. Again, bibliographic coupling puts this research in its two largest clusters 1 and 2. Other unique lines of inquiry can be made out, too (e. g., cluster 10 on the role of solar winds in the sun's heliosheath), but ultimately require the expert interpretation of astronomers.

**Table 8. Top-ten tf-idf terms for bibliographic coupling in astronomy.**

| Cluster | Top-ten tf-idf terms |
|---|---|
| 1 | ray, galaxies, gas, emission, galaxy, star, black, disk, mass, accretion |
| 2 | galaxies, star, stars, stellar, disk, galaxy, emission, mass, ray, gas |
| 3 | ray, disk, star, galaxies, emission, stars, gas, stellar, mass, galaxy |
| 4 | stars, star, emission, gas, mass, lines, stellar, binary, ray, formation |
| 5 | magnetic, solar, coronal, flare, cme, reconnection, region, flux, field, plasma |
| 6 | frb, ray, radio, emission, frbs, pulsar, bursts, magnetic, neutron, pulsars |
| 7 | magnetic, solar, ibex, wind, turbulence, plasma, interstellar, reconnection, field, electron |
| 8 | dust, planet, comet, disk, surface, au, solar, planets, belt, asteroids |
| 9 | solar, magnetic, black, stars, star, mass, galaxies, formation, stellar, gas |
| 10 | galaxies, galaxy, radio, â, telescope, greenburst, ost, bispectrum, survey, sky |

**Table 9. Cluster network measures for publication authorship in astronomy (10 largest clusters).**

| Cluster | Size | Age (mean) | Age (s.d.) | Degree | Betweenness | Closeness |
|---|---|---|---|---|---|---|
| 1 | 1009 | 5.10 | 2.80 | 40 | 345.58 | 0.01 |
| 2 | 978 | 5.74 | 2.90 | 31 | 38.83 | 0.01 |
| 3 | 964 | 5.72 | 2.83 | 29 | 41.58 | 0.01 |
| 4 | 863 | 4.93 | 2.78 | 37 | 271.73 | 0.01 |
| 5 | 761 | 5.55 | 2.99 | 33 | 139.12 | 0.01 |
| 6 | 729 | 5.78 | 2.87 | 30 | 79.89 | 0.01 |
| 7 | 683 | 5.73 | 2.94 | 33 | 82.18 | 0.01 |
| 8 | 602 | 5.29 | 2.84 | 31 | 84.49 | 0.01 |
| 9 | 579 | 5.49 | 2.97 | 30 | 36.58 | 0.01 |
| 10 | 577 | 5.58 | 2.97 | 28 | 29.89 | 0.01 |

**Table 10. Top-ten tf-idf terms for publication authorship in astronomy (10 largest clusters).**

| Cluster | Top-ten tf-idf terms |
|---|---|
| 1 | magnetic, solar, coronal, flare, reconnection, loops, flux, line, field, cme |
| 2 | emission, dust, disk, gas, molecular, star, stars, maser, observations, infrared |
| 3 | galaxies, star, galaxy, gas, redshift, emission, infrared, dust, formation, mass |
| 4 | sn, sne, ia, optical, star, supernova, explosion, emission, mass, ray |
| 5 | planet, planets, kepler, stars, stellar, transit, transiting, star, exoplanet, planetary |
| 6 | stars, galaxies, stellar, clusters, star, cluster, fe, galaxy, globular, mass |
| 7 | ray, radio, $\gamma$, emission, pulsar, gamma, fermi, telescope, flux, source |
| 8 | radio, galaxies, wind, turbulence, array, field, magnetic, galaxy, emission, survey |
| 9 | ray, black, kev, hole, accretion, neutron, emission, star, outburst, source |
| 10 | ibex, solar, wind, magnetic, voyager, heliosheath, plasma, interstellar, field, spacecraft |

**Gastroenterology.** Bibliographic coupling in gastroenterology presents as a dense network of 26 connected clusters. The ten largest clusters make up a little more than 80% of all articles. The periphery is negligible with no more than 21 articles found in the seven smallest clusters (Table 11). Gastroenterology is dominated by Latin terminology and medical abbreviations foreign to laypersons (Table 12). Examples for research foci in gastroenterology include Crohn's disease (cluster 1), liver cirrhosis (cluster 2), colorectal cancer (crc; cluster 4).

**Table 11. Network measures for bibliographic coupling in gastroenterology.**

| Cluster | Size | Age (mean) | Age (s.d.) | Degree | Betweenness | Closeness |
|---|---|---|---|---|---|---|
| 1 | 2352 | 5.44 | 2.88 | 20 | 56.97 | 0.03 |
| 2 | 2132 | 5.58 | 3.02 | 18 | 30.87 | 0.03 |
| 3 | 1257 | 6.31 | 2.97 | 17 | 7.53 | 0.03 |
| 4 | 1168 | 5.56 | 2.98 | 20 | 76.30 | 0.03 |
| 5 | 819 | 6.10 | 2.94 | 18 | 9.97 | 0.03 |
| 6 | 817 | 4.27 | 2.44 | 12 | 0.43 | 0.03 |
| 7 | 553 | 5.53 | 3.11 | 18 | 9.97 | 0.03 |
| 8 | 474 | 5.77 | 3.02 | 17 | 28.68 | 0.03 |
| 9 | 237 | 5.81 | 2.92 | 13 | 1.77 | 0.03 |
| 10 | 213 | 6.73 | 2.83 | 15 | 2.52 | 0.03 |

**Table 12. Top-ten tf-idf terms for bibliographic coupling in gastroenterology.**

| Cluster | Top-ten tf-idf terms |
|---|---|
| 1 | ibd, cd, uc, crohn's, disease, colitis, remission, patients, anti, infliximab |
| 2 | hcv, liver, hcc, hbv, nafld, hepatitis, cirrhosis, svr, fibrosis, patients |
| 3 | gastric, pylori, reflux, gerd, oesophageal, cells, barrett's, eac, esophageal, ppi |
| 4 | crc, colonoscopy, colorectal, cancer, screening, risk, ci, adenomas, fit, polyps |
| 5 | ibs, symptoms, constipation, placebo, pain, rome, irritable, bowel, symptom, patients |
| 6 | gastric, cancer, survival, gastrectomy, patients, gc, chemotherapy, esd, metastasis, os |
| 7 | psc, aip, mice, pbc, microbiota, liver, cdi, fmt, gut, udca |
| 8 | pancreatic, pancreatitis, mice, pdac, cells, acinar, ap, cp, acute, kras |
| 9 | gluten, celiac, coeliac, cd, ttg, disease, gfd, hla, gliadin, biopsy |
| 10 | bleeding, ulcer, aspirin, risk, rebleeding, peptic, ci, ugib, ulcers, users |

Publication authorship in gastroenterology expands the number of connected clusters from 26 to 37 (Table 13). This more detailed picture is best exemplified with cancer research in gastroenterology. Bibliographic coupling groups gastric and colorectal (crc) cancer into clusters 4 and 6. In contrast, publication authorship clearly shows the four most common types of gastrointestinal cancers. First and second, gastric cancer (clusters 1 and 8) and colorectal cancer (cluster 6) are immediately visible as distinct fields of interest. Moreover, colorectal cancer often coincides with inflamatory bowel disease (ibd) and eosinophilic esophagitis (eoe), both of which are large parts of cluster 5. Liver cancer (clusters 1, 3, and 8) and pancreatic cancer (cluster 9) mark the third and fourth most common type of cancer.

The distribution of the most common types of gastrointestinal cancer across clusters finds explanation in additional tf-idf terms that relate to common practice in treatment or diagnosis. For example, cluster 1 highlights endoscopic submucosal dissection (esd) as preferential treatment of gastric or liver cancer in patients. In contrast, cluster 8 puts forward diagnostic research on the expression and risk of early gastric cancer (Table 14).

The development of the scientific frontier is not immediately apparent for publication authorship, although the more detailed picture allows even a layperson to make out clear distinctions within research sub-fields such as the focus on treatment of gastric or liver cancer in cluster 1 and 3 as opposed to the diagnosis of these types of cancer in cluster 8. Further interpretations that may shed a light on the latest developments in the research field call for the expertise of gastroenterologists.

**Table 13. Network measures for publication authorship in gastroenterology (10 largest clusters).**

| Cluster | Size | Age (mean) | Age (s.d.) | Degree | Strength | Betweenness | Closeness |
|---|---|---|---|---|---|---|---|
| 1 | 1707 | 5.63 | 2.90 | 18 | 16.03 | 0.02 | |
| 2 | 1608 | 5.45 | 2.99 | 29 | 201.39 | 0.02 | |
| 3 | 1414 | 5.47 | 3.01 | 26 | 145.39 | 0.02 | |
| 4 | 1207 | 5.36 | 2.94 | 23 | 63.94 | 0.02 | |
| 5 | 970 | 5.54 | 2.88 | 22 | 69.64 | 0.02 | |
| 6 | 967 | 5.52 | 2.94 | 17 | 5.44 | 0.02 | |
| 7 | 919 | 6.01 | 2.90 | 20 | 35.81 | 0.02 | |
| 8 | 590 | 5.13 | 2.85 | 16 | 2.88 | 0.02 | |
| 9 | 422 | 5.84 | 2.97 | 16 | 7.70 | 0.02 | |
| 10 | 287 | 5.59 | 2.81 | 15 | 8.49 | 0.02 | |

**Table 14. Top-ten tf-idf terms for publication authorship in gastroenterology (10 largest clusters).**

| Cluster | Top-ten tf-idf terms |
|---:|---|
| 1 | patients, cancer, gastric, hcc, liver, japanese, esd, cells, mice, hcv |
| 2 | liver, hcv, patients, cirrhosis, hepatitis, hcc, fibrosis, risk, ci, hbv |
| 3 | mice, cells, liver, hcc, expression, cancer, patients, cell, tissues, mir |
| 4 | cd, patients, ibd, uc, remission, infliximab, crohn's, anti, week, colitis |
| 5 | ibd, patients, cd, ci, risk, eoe, children, cancer, uc, disease |
| 6 | crc, patients, screening, colorectal, fit, colonoscopy, risk, ci, cancer, ibd |
| 7 | ibs, symptoms, patients, reflux, symptom, placebo, pain, gerd, constipation, oesophageal |
| 8 | gastric, cancer, patients, cells, mice, liver, hcc, expression, ci, risk |
| 9 | pancreatic, pancreatitis, mice, patients, cells, ap, cancer, acinar, cp, pdac |
| 10 | ibd, patients, gluten, cd, ifx, uc, anti, disease, liver, ci |

## Discussion

Publication authorship proves a point in displaying a more detailed picture of research than bibliographic coupling. Of course, it is only one methodological approach among many others used in bibliometric studies. Alternatives to bibliometric networks based on (co-)citation or author(ship) include the co-occurrences of words in the title or abstract of articles and topic modeling algorithms such as, for example, Latent Dirichlet Allocation (LDA). While bibliometric networks do not immediately compare to these alternative approaches, I discuss some findings from running a cluster analysis of word co-occurrences in abstracts as well as an LDA for the research field of accounting.

### Co-word analysis

Co-word analysis [32, 33] looks at the intellectual organization of research based on the co-occurrences of article keywords. Its strength is a simple setup of words as vertices and edges as their co-occurrences, commonly weighted by an equivalency index [34] similar to the above discussed measure of term frequency-inverse document frequency (tf-idf). However, a first trouble with co-word analysis is that not all articles in scientific databases come with keywords, mostly due to the fact that some journals do not require authors to supply keywords to their articles. This trouble shows most prominently when approximately one third of all articles in accounting and more than 40 percent of all articles in gastroenterology have no associated keywords. It is somewhat less of a concern in astronomy where only five percent of all articles are missing keywords.

In order to have the same baseline number of articles as the above studies in bibliographic coupling and publication authorship, I use words in article abstracts instead of keywords to compute word co-occurrences. I run the same network statistics and cluster analysis in the co-word analysis of accounting, astronomy, and gastroenterology in order to highlight similarities and differences to bibliographic coupling and publication authorship. Most notably, the number of vertices and edges increases dramatically in co-word analysis now that words instead of articles are the starting point (Table 15). Density, transitivity, and assortativity hover around the same values, though they defy any immediate comparison among the disparate networks.

The number of clusters steadily increases from accounting to astronomy to gastroenterology (Fig 5). At first sight, it appears as if co-word analysis provides a segmentation of research somewhat opposite to publication authorship where the number of clusters decreases. However, the distribution of clusters reveals that all three academic areas feature one huge cluster

**Table 15. Network measures for co-word analysis.**

|  | Accounting | Astronomy | Gastroenterology |
|---|---|---|---|
| Vertices | 16791 | 38180 | 52481 |
| Isolates | 0 | 0 | 0 |
| Edges | 3679818 | 11151688 | 18026827 |
| Density | 0.03 | 0.02 | 0.01 |
| Transitivity | 0.27 | 0.25 | 0.20 |
| Assortativity | -0.28 | -0.36 | -0.32 |
| Clusters | 96 | 146 | 212 |

of words that are most common to all articles. Disregarding this pool cluster, we observe a more even distribution of words among clusters. While these word clusters describe research fields in great detail, a second and major drawback of co-word analysis is that words are exclusive to clusters. A fairly common word in accounting such as stocks, for example, necessarily appears only in single cluster then. This calls into question the meaningfulness of clusters in

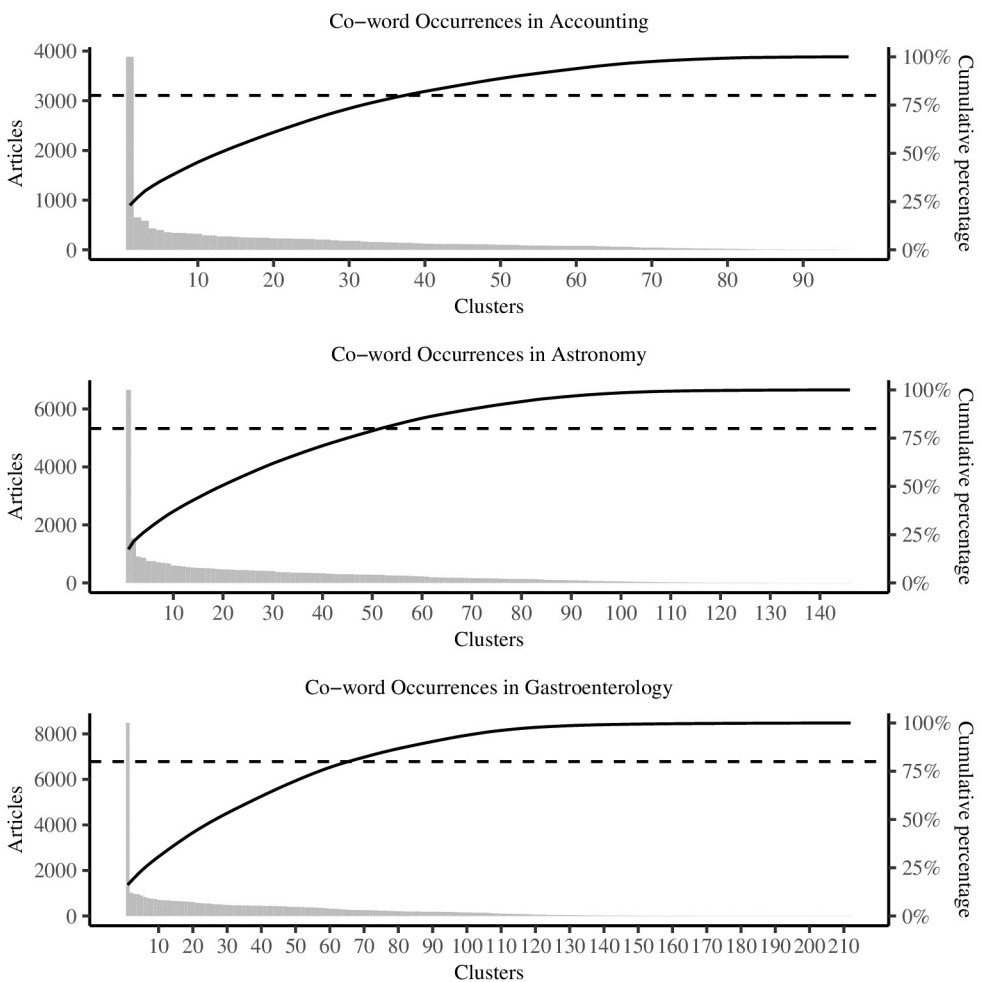

**Fig 5. Cluster distribution in co-word analysis.**

the first place. Bibliographic coupling and publication authorship both provide a first layer of connectivity among articles that in later analysis allows for terms to appear in multiple, overlapping clusters, which is much better suited to describe the segmentation of research and the development of the scientific frontier.

### Latent Dirichlet Allocation

An alternative to the exclusive clusters of co-word analysis is Latent Dirichlet Allocation (LDA) [35]. LDA is a generative probabilistic model based on the idea that each document (e. g., an article abstract) in a corpus is a random mix of latent topics, and each topic is in turn characterized by a probability distribution over words. For example, the terms stocks and economy are both likely to make up a topic that describes the impact of stocks on a country's economy (e. g., Novo Nordisk's market value has now exceeded the size of the entire Danish economy), whereas they are perhaps less likely to appear in a topic that outlines the connection between the initial public stock offering (IPO) and the economy (e. g., California frequently has a large budget surplus due to income taxes of IPO sales).

While LDA yields topics similar to the clusters of bibliometric coupling and publication authorship, it shares little commonalities with the network analysis of vertices and edges. Its biggest drawback is that the number of topics needs to be fixed a priori, though there are several ways to determine the optimal number of topics by now [36–38]. Another weak spot is that it requires significant computational power. Indeed, computing the optimal number of topics in 25 iterations of LDA in accounting failed due to issues of memory allocation on the ten cores of an Apple Silicon M1 Max with 32 GB RAM. The following computations were instead carried out on 64 Intel Xeon high-performance cores with 364 GB RAM. Running time was around 35 minutes for accounting, 74 minutes for astronomy, and 116 minutes for gastroenterology. In contrast, the entire computations in bibliographic coupling and publication authorship run in less than three minutes on Apple Silicon for all three academic areas combined.

Fig 6 shows normalized values for a number of topics ranging from 10 to 250 in accounting, astronomy, and gastroenterology. With a look for either a minimal [37, 38] or a maximal [36] value, the optimal number of topics falls somewhere between 70 and 140 in accounting, between 100 and 140 in astronomy, and between 100 and 160 topics in gastroenterology.

Already the number of topics at the lower end of the range for each academic area is larger than then number of connected clusters in bibliographic coupling and publication authorship, which suggests a greater detail of research segmentation. However, LDA offers limited information on the organization of research beside the document-topic probability $\gamma$ and the topic-word probability $\beta$. On the one hand, $\gamma$ indicates the probability with which a topic represents a document; on the other hand, $\beta$ indicates the probability with which a word is common to a topic. Taken together, Table 16 shows the top-ten topics in decreasing order of their mean $\gamma$ alongside the respective top-ten terms in decreasing order of their $\beta$ scores.

While some topics in LDA compare favorably to clusters in bibliographic coupling (e. g., topic 63 and cluster 4 on knowledge and innovation) and publication authorship (e, g., topic 54 and cluster 8 on the role of analysts in firm earnings or topic 40 and cluster 9 on the quality of audits), others certainly require their own interpretation (e. g., topic 22 on accounting practices and accountability). Unfortunately, additional information on centrality, size, or age of topics similar to clusters is not readily available in LDA. The segmentation of research by topics in LDA is perhaps similar to the one by cluster, though the development of the scientific frontier is not easy to spot, not least because of the missing information on the organization of research areas.

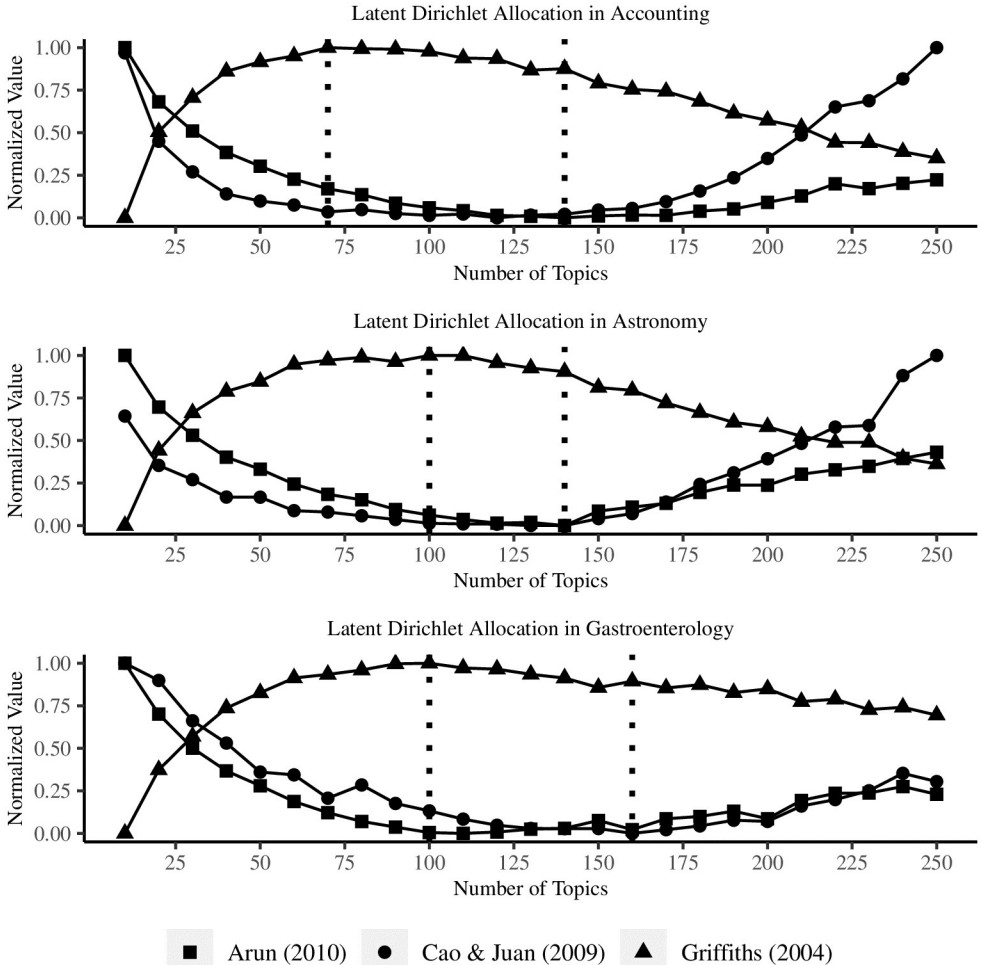

**Fig 6. Topic ranges for Latent Dirichlet Allocation.**

**Table 16. Top-ten topics and terms for Latent Dirichlet Allocation in accounting.**

| Topic | Top-10 Beta Terms |
|---|---|
| 64 | returns, stocks, return, stock, expected, factor, market, average, momentum, pricing, portfolio |
| 22 | accounting, paper, practices, practice, study, professional, process, processes, accountability, understanding |
| 67 | model, asset, equilibrium, pricing, develop, consumption, dynamic, time, prices, run |
| 38 | banks, bank, loan, credit, loans, crisis, lending, borrowers, lenders, banking |
| 18 | research, university, academic, universities, entrepreneurial, knowledge, entrepreneurship, scientists, spin, development |
| 54 | analysts, news, earnings, forecasts, analyst, forecast, announcements, bad, announcement, coverage |
| 63 | innovation, knowledge, relationship, projects, innovative, technological, research, activities, paper, collaboration |
| 40 | audit, quality, auditors, auditor, client, fraud, fees, clients, expertise, auditing |
| 26 | firms, firm, low, larger, consistent, u.s., affect, matching, propensity, firm's |
| 1 | trading, trade, short, traders, sales, informed, trades, selling, volume, insider |

## Conclusion

Bibliometric studies are common practice in all academic disciplines. They assess the history of a research field, point out the state of the art, and identify the development of the scientific frontier. Bibliometric studies are transparent, reproducible, and scalable, making them a cost-effective way of analyzing large volumes of academic articles. In the end, they highlight idio-syncrasies of scientific work that are insightful to both laypersons and experts.

From classic approaches of mapping research publications by (co-)citation and biblio-graphic coupling to centering on collaboration among scholars by author co-citation, author bibliographic coupling, and co-authorship, the methodology of bibliometric studies has gotten more and more technically refined. Still, there are some limitations. For example, (co-)citation analysis and bibliographic coupling do not capture the reasoning behind citations. Whether articles are cited to make or break an argument is therefore unknown. Publication authorship does away with this limitation by accounting for both the social dimension of authorship and the intellectual dimension of scientific work.

Analyzing the content of academic articles, of course, is the prime domain of natural language processing. The findings of bibliometric studies may thus be further interpreted using measures such as term frequency-inverse document frequency (tf-idf) to highlight scientific concepts that are most descriptive for academic areas. Together with measures on the level of vertices and edges (e. g., degree, betweenness, closeness, size, age) and on the level of the bib-liometric network in question (e. g., density, assortativity, transitivity), the segmentation of research becomes not only more interpretable but also comparable across the space and time of scientific work.

Of course, bibliometric studies are far from the only means of inquiry into the segmentation of research and the development of the scientific frontier. Approaches used in natural language processing such as, for example, the analysis of word co-occurrences and Latent Dirichlet Allo-cation (LDA) are particularly suited to capture the intellectual dimension of scientific work without necessarily inheriting the limitations of (co-)citation analysis and other bibliometric approaches. However, they are computationally costly to begin with and their findings are often harder to interpret without the backdrop of additional measures from the realm of bib-liometric studies.

The key differences between publication authorship and approaches in natural language processing such as LDA are what makes bibliometric studies attractive in the first place. Publi-cation authorship is transparent in both its definition of what vertices and edges are and its analysis of the respective bibliometric networks. It is easily reproducible not only across the space of multiple disciplines but also across the time of a single discipline, which allows for a comparison of different academic areas and an interpretation of the development of the scien-tific frontier. Last, publication authorship scales well from small fields of research to large vol-umes of academic articles. In contrast, LDA as a generative probabilistic model is somewhat opaque, not least because it requires the specification of the number of clusters and a number of training parameters to begin with. Its findings are also more difficult to interpret without additional measures derived from the structure of scientific work. And it is computationally intense, which makes it a costly alternative to bibliometric studies.

Consider that publication authorship clearly identifies themes and topics in accounting despite the lower number of clusters. For example, one cluster shows a large but rather periph-eral body of work on international financial reporting standards, whereas another cluster that comprises of a slightly smaller number of academic articles on high-frequency trading systems sits in the center of adjacent work in accounting. LDA is more generic, despite the fact that its higher number of clusters suggest more detail. For example, it shows a cluster about a cluster

about banking and credit, a cluster about innovation and research, and a cluster about trading and insider information. None of these clusters are immediately identifiable as larger or smaller, central or peripheral, older or younger.

Admittedly, the latest developments in artificial intelligence promise to remedy some of these shortcomings in natural language processing (e. g., ChatGPT-4 suggests that Habermas and Luhmann are intellectual rivals despite the fact that they published together; at the same time, it cannot correctly identify the DOIs of either works). Unfortunately, artificial intelligence with hundreds of billions of parameters or more operates largely as a black box. Perhaps there is still room for bibliometric studies carefully rooted in theory then.

Publication authorship, I argue, offers a more fine-grained picture of academic research that provides explanatory power beyond simple refinement. The illustrations of bibliographic coupling versus publication authorship in accounting, astronomy, and gastroenterology ultimately confirm significant benefits to bibliometric studies of scientific work.

Moreover, the idea to connect publications by authorship immediately extends to, for example, organization studies. Following the now popular notion that communication constitutes organization [39–41], we may conceive of corporate documents such as meeting minutes, project reports, or product presentations as communication episodes [21]. The authorship of these episodes, in turn, provides the proverbial glue among the said documents. Documents and authorship are therefore conceived as the vertices and the edges that map out an organization as a network of communication episodes. A respective cluster analysis commonly shows the functions of an organization (e. g., accounting, engineering, marketing) similar to the subfields of an academic discipline [21]. Indeed, an academic discipline may well be thought of as an organization of the scientific work conducted within the disciplinary boundaries. My hope then is that publication authorship provides another useful approach in the toolbox of bibliometric studies and beyond.

## Supporting information

**S1 Appendix. Top-10 journals in accounting, astronomy, and gastroenterology.**
(PDF)

**S2 Appendix. Glossary.**
(PDF)

## Author Contributions

**Conceptualization:** Steffen Blaschke.

**Data curation:** Steffen Blaschke.

**Formal analysis:** Steffen Blaschke.

**Investigation:** Steffen Blaschke.

**Methodology:** Steffen Blaschke.

**Project administration:** Steffen Blaschke.

**Resources:** Steffen Blaschke.

**Software:** Steffen Blaschke.

**Supervision:** Steffen Blaschke.

**Validation:** Steffen Blaschke.

**Visualization:** Steffen Blaschke.

**Writing – original draft:** Steffen Blaschke.

**Writing – review & editing:** Steffen Blaschke.

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
