## [Decision Letter · Decision Letter 0]

20 Jan 2023

PONE-D-22-17286Publication authorship: A new approach to the bibliometric study of scientific work and beyondPLOS ONE

Dear Dr. Blaschke,

Thank you for submitting your manuscript to PLOS ONE. After careful consideration, we feel that it has merit but does not fully meet PLOS ONE’s publication criteria as it currently stands. Therefore, we invite you to submit a revised version of the manuscript that addresses the points raised during the review process.

ACADEMIC EDITOR: One reviewer is quite detailed, the other one is perhaps too general. Nevertheless, please, provide a careful reply to all the comments. The paper has to be substantially revised. You should consider the present one as a risky major revision.

We look forward to receiving your revised manuscript.

Kind regards,

Roy Cerqueti, Ph.D.

Academic Editor

PLOS ONE

Journal Requirements:

Reviewers' comments:

Reviewer's Responses to Questions

**Comments to the Author**

1. Is the manuscript technically sound, and do the data support the conclusions?

Reviewer #1: Yes

Reviewer #2: Yes

2. Has the statistical analysis been performed appropriately and rigorously? 

Reviewer #1: I Don't Know

Reviewer #2: Yes

3. Have the authors made all data underlying the findings in their manuscript fully available?

Reviewer #1: Yes

Reviewer #2: Yes

4. Is the manuscript presented in an intelligible fashion and written in standard English?

Reviewer #1: Yes

Reviewer #2: Yes

5. Review Comments to the Author

Reviewer #1: This manuscript introduced publication authorship as a complement to bibliometric study.

Three data sets of academic articles from accounting, astronomy, and gastroenterology are used to illustrate the benefits of publication authorship.

However, in my opinion, the content of this paper lacks sufficient attraction. Using publication authorship as the connection point of the literature coupling network might improve the clustering quality, but the author should first provide a feasible practical solution for other researchers to validate. In addition, the figures and tables used in the paper do not effectively help readers understand the author's viewpoints. The terms used in the tables such as age, strength, and degree are not clear enough for readers to understand what the meaning is, and the figures are not clear enough.

Reviewer #2: This paper introduces a novel approach to investigate bibliometric networks, i.e., what the author calls "publication authorship". He shows that publication authorship allows to identify different research topics more granularly than bibliographic coupling, being able to distinguish between contiguous research fields. However, there are some issues that, in my opinion, need to be fixed in order to enhance the contribution of this work.

1. The whole paper is a bit unstructured, in the sense that there is not a straightforward storytelling. For instance, once I finished reading the "clustering" section on page 8, I felt as I was at the end of the paper, and I did not know what to expect in the next pages. It would be useful to introduce your research design before going through the analysis of the case study, either as an independent section, or as a part of an existing section, e.g., the introduction.

2. The description of the theoretical background is scarce. The other bibliometric approaches (i.e., co-citation, bibliographic coupling, author co-citation, bibliographic coupling, and co-authorship) should be discussed in a more extensive way, referring to a wide list of reference works in the literature.

3. In relation to the previous comment, I think co-word and topical networks should be considered as well. In particular, I suggest comparing the results obtained through publication authorship not only with bibliographic coupling, but also with co-citation, co-word and topical networks. Specifically, the last two methods are used to identify topic similarity between publications by mining words contained in titles or abstracts. Thus, it would be highly relevant to assess if publication authorship provides similar or different results compared to co-word and topical networks, and if it confirms being more (or as) granular than these two other methods in the identification of research fields and topics.

4. The analysis of the different streams, i.e., accounting, astronomy, and gastroenterology is very detailed. However, I suggest the author comment also on the differences between the three areas. For instance, are there specific behaviors in each research field that motivate higher/lower density, transitivity, assortativity, and so on? For instance, the average number of publications of a scholar in each area may affect these results. Or again, is one area more clustered than the others by definition?

5. It could be appropriate to define a sort of "glossary of network metrics and properties" rather than introducing them one by one within the text.

6. I think there are two typos in Table 5 and Table 6. The density values corresponding to publication authorship should be 0.001 and 0.004, respectively.

I hope these comments will help the author improve his contribution.

6. PLOS authors have the option to publish the peer review history of their article (what does this mean?). If published, this will include your full peer review and any attached files.

Reviewer #1: No

Reviewer #2: No

---

## [Author Response · Author response to Decision Letter 0]

14 Apr 2023

Dear Roy: Thank you for your guidance on how to tackle the issues raised by the reviewers. I have been closely following your suggestions. You will find my responses to the reviewers below.

Reviewer #1

Dear Reviewer #1: Thank you for your comments on my manuscript. In the following, I will take upon each one of them and offer revisions accordingly.

Practical Solution

You start by saying that the “paper lacks sufficient attraction. Using publication authorship as the connection point of the literature coupling network might improve the clustering quality, but the author should first provide a feasible practical solution for other researchers to validate.”

The most practical solution I can provide is access to the source code my analysis is based on. Both the code as well as a step-by-step research protocol of how to obtain the data sets I used in my illustrations are available on GitHub (cf. reference on p. 4). The protocol may easily be adapted to curate data sets of your own interest, of course.

Tables

You continue with the observation that “the figures and tables used in the paper do not effectively help readers understand the author’s viewpoints. The terms used in the tables such as age, strength, and degree are not clear enough for readers to understand what the meaning is.”

I value this observation, but I must ask you to specify what you are looking for. The paper describes common measures in network analysis in layperson’s terms (e.g., saying that the degree is the number of edges a vertex has to other vertices, cf. p. 9) as well as points to the appropriate references on how these measures are used in the literature (e. g., Leydesdorff on betweenness as an indicator of interdisciplinarity, cf. p. 9). One issue that came about your observation is that age was not identified with its units. I corrected this shortcoming, and now age is given in years.

Figures

You end on the comment that “the figures are not clear enough.”

I agree with you that some figures are of low quality (i.e., they appear blurry). The issue only occurred when the PLOS ONE submission engine stitched together the manuscript and the separately uploaded figures. The fig- ures are high-quality PostScripts to begin with, so I managed a workaround by including them as attachments to the manuscript myself. The issue is thus

hopefully resolved.

Reviewer #2

Dear Reviewer #2: Thank you for your comments on my manuscript. In the following, I will take upon each one of them and offer revisions accordingly.

Structure

You say that “the paper is a bit unstructured” and already ask to “introduce [the] research design before going through the analysis of the case study.”

I am glad that you bring this issue up. The introduction now includes an overview of the research design. I hope by this you and the readers do not get lost in the technical details but remain on track following the research step by step.

Theoretical Background

“The description of the theoretical background is scarce,” you mention. In this connection, you ask that the briefly introduced bibliographic approaches should be discussed in a more extensive way.

I am happy to accommodate this suggestions. The theoretical considera- tions are now divided into three subsections (on the intellectual structure, the social structure, and the combination of both these structure of scientific work). The considerations fill in details on each one of the five established bibliometric approaches (i.e., co-citation, bibliographic coupling, author co-citation, bibli- ographic coupling, and co-authorship) including a larger list of references to existing literature. I hope that this satisfies your call for a more broader theo- retical background in the sense of a brief literature review. I refrain from going into more detail since in my humble opinion this would unbalance the intro- ductory part of the paper opposite the empirical illustrations and statistical analysis.

Co-word and Topic Networks

You argue that “co-word and topical networks should be considered as well” and readily suggest to compare “the results obtained through publication authorship not only with bibliographic coupling, but also with co-citation, co-word and topical networks.”

I fully agree with you that “it would be highly relevant to assess if publica- tion authorship provides similar or different results compared to co-word and topical networks.” However, there is a simple reason why I cannot provide a full-fledged analysis of co-citation, co-word, and topic networks alongside bibli- ographic coupling and publication authorship. PLOS ONE does not explicitly limit the number of pages a paper can have, but it encourages a concise and accessible writing style. My analysis and discussion of bibliographic coupling opposite of publication authorship already make up more than half the paper. I estimate that an analysis of co-citation, co-word, and topic networks would add at least eight to ten pages. Next to an increase in the length of the pa- per, its readability would likely suffer from a comparison between publication authorship and now four (instead of one) other bibliometric approaches.

I am therefore not providing a full-fledged analysis of co-citation, co-word, and topic networks. Instead, you will find a more theoretical discussion of bibliometric networks based on word co-occurrences alongside an outlook of further work in the conclusion.

Commonalities and Differences Between Fields of Research

You point out that the analysis of the three academic areas (accounting, as- tronomy, and gastroenterology) is very detailed, yet it lacks comments on the differences between them. You specifically suggest to look into “specific be- haviors in each research field that motivate higher/lower density, transitivity, assortativity, and so on.”

Again, thank you for these insightful suggestions, which I gladly take up. Of course, you are absolutely right that each academic discipline has its unique features (e.g., a lower average number of authors on papers in accounting than in gastroenterology, mentioned on p. 5). This then immediately reflect in the findings such as a different densities of bibliometric networks. My empirical illustrations, however, pit bibliographic coupling against publication authorship across disciplines. They are not meant to be comparing academic areas to one another.

To take up your point, I have now included some clarifying remarks at the beginning of the data section. I hope this will satisfy your curiosity of differences between and among fields of research for now.

Glossary

“It could be appropriate to define a sort of ‘glossary of network metrics and properties’ rather than introducing them one by one within the text,” you men- tion.

Yes, perhaps it is a good idea to include a glossary for the reader. I will consult with the editor in what way he thinks a glossary is appropriate to include.

Tables

You point out “two typos in Table 5 and Table 6. The density values corre- sponding to publication authorship should be 0.001 and 0.004, respectively.”

Thank you for doing so. You are, of course, correct. The reported values should have been and now do read “< 0.01” (because PLOS ONE asks for only two decimal places).

---

## [Decision Letter · Decision Letter 1]

16 May 2023

PONE-D-22-17286R1Publication authorship: A new approach to the bibliometric study of scientific work and beyondPLOS ONE

Dear Dr. Blaschke,

Thank you for submitting your manuscript to PLOS ONE. After careful consideration, we feel that it has merit but does not fully meet PLOS ONE’s publication criteria as it currently stands. Therefore, we invite you to submit a revised version of the manuscript that addresses the points raised during the review process.

We look forward to receiving your revised manuscript.

Kind regards,

Roy Cerqueti, Ph.D.

Academic Editor

PLOS ONE

Reviewers' comments:

Reviewer's Responses to Questions

**Comments to the Author**

1. If the authors have adequately addressed your comments raised in a previous round of review and you feel that this manuscript is now acceptable for publication, you may indicate that here to bypass the “Comments to the Author” section, enter your conflict of interest statement in the “Confidential to Editor” section, and submit your "Accept" recommendation.

Reviewer #2: (No Response)

2. Is the manuscript technically sound, and do the data support the conclusions?

Reviewer #2: Yes

3. Has the statistical analysis been performed appropriately and rigorously? 

Reviewer #2: Yes

4. Have the authors made all data underlying the findings in their manuscript fully available?

Reviewer #2: Yes

5. Is the manuscript presented in an intelligible fashion and written in standard English?

Reviewer #2: Yes

6. Review Comments to the Author

Reviewer #2: Dear author, thanks for this revised version of your paper. As I told you in my previous comments, your paper introduces a very interesting approach to investigate bibliometric networks, and your statistical analysis is also well developed. The new paragraph in the Introduction section is definitely useful to provide the reader with an overview about your research design. However, some of the other points raised during the previous round of review were not met properly in this revised version of your work. Thus, I will try to explain them better in this second review.

1. The theoretical background of the different bibliometric approaches still remains scarce. You expanded the part devoted to the description of the approaches, but what I meant in my previous round of comments is that a literature review on existing works in which the methods are described and applied is essential. This would help clarify when one approach is more relevant than the others, and which are the gaps that publication authorship aims to fill in such a framework.

2. In relation to my comment about the analysis of co-word and topical networks, you argue that the length (and the readability) of the paper would suffer from this integration, and on page 17 (line 510) you just say that "I estimate it to be considerably closer to clusters found using co-word analysis. The extent to what this estimate is true is certainly worthwhile to explore in future scientific work". Unfortunately, this answer is not enough. I understand that it may be difficult to compare publication authorship to all the other methods. Nonetheless, I consider necessary to analyze at least one among topical and co-word networks. This is also in line with my previous comment. You need to point out which is the gap in the literature that your method addresses. If you are providing something that is able to identify research fields and topics more granularly, you must demonstrate it through the comparison with one of the two aforementioned methods (since they are the bibliometric approaches that are "alternative" to yours). Then, you can leave all the other analyses as future developments.

3. I will wait for a glossary of network metrics and properties to be included in the revised version of your work.

I hope my comments are clearer in this second round of review.

7. PLOS authors have the option to publish the peer review history of their article (what does this mean?). If published, this will include your full peer review and any attached files.

Reviewer #2: No

---

## [Author Response · Author response to Decision Letter 1]

1 Nov 2023

Dear Reviewer #2: Thank you for your extended comments on my manuscript. In the following, I will take upon each one of them and offer revisions accordingly.

Theoretical Background

Following your argument that “literature review on existing works in which the methods are described and applied is essential,” I now discuss highly-cited research on the intellectual and social structure of scientific work. While this brief review of existing works is necessarily selective, I also include references to comprehensive reviews of the theory and practice of bibliometric studies. My hope is the examples of previous work and the pointer to reviews furthers my argument that publication authorship is a valid alternative to mainstay methods in bibliometrics.

Topic and Co-word Networks

You “consider it necessary to analyze at least one among topical and co-word networks” in order to compare it to bibliometric coupling and publication authorship. I am happy to report that the manuscript now includes not just one but two alternative methods in the discussion.

First, I ran a co-word analysis on words in the abstracts of articles in accounting, astronomy, and gastroenterology. The results of the analysis are unsurprisingly not as pronounced as the more granular picture that publication authorship delivers, mostly due to the fact that co-word analysis necessarily yields mutually exclusive word clusters.

Second, I ran Latent Dirichlet Allocations (LDAs) for the three academic areas in question. The results are in line with publication authorship if not somewhat more refined as the now included example in accounting shows. How- ever, there are two major drawbacks to running LDAs. On the one hand, they require the researcher to fix the number of clusters a priori, which always entails a particular level of arbitrariness. On the other hand, the underlying machine learning algorithm of LDAs forced me to switch from my personal computer to a high-performance computing platform in the cloud where computations still ran for hours.

I hope that this newly included discussion of alternative methods in bibliometric studies satisfies your curiosity.

Glossary

As per your suggestion, the revised manuscript now includes a glossary of the most prominent concepts.

Additional Refinements

The second revision of the manuscript includes several refinements. Most importantly, an update to one of the R packages I relied on earlier prompted me rewrite my own code. In the process of doing so I had to discover that this pre- viously used package roughly doubles the values for the basic document-term matrix for publication authorship, whereupon inflated values made it into the original reporting of the results.

After double (and triple, and quadruple) checking my own code, I can safely report that all reported values still yield the same implications. (After all, the previously used package merely doubled the values, which does not change much in the relation among numbers.) In the course of action, I have streamlined the discussion of the findings in accounting, astronomy, and gastroenterology. Hopefully the manuscript is now a more straight-forward read, not least because it quicker makes it to the discussion of alternative methods in bibliometric studies.

---

## [Decision Letter · Decision Letter 2]

30 Nov 2023

PONE-D-22-17286R2Publication authorship: A new approach to the bibliometric study of scientific work and beyondPLOS ONE

Dear Dr. Blaschke,

Thank you for submitting your manuscript to PLOS ONE. After careful consideration, we feel that it has merit but does not fully meet PLOS ONE’s publication criteria as it currently stands. Therefore, we invite you to submit a revised version of the manuscript that addresses the points raised during the review process.

We look forward to receiving your revised manuscript.

Kind regards,

Roy Cerqueti, Ph.D.

Academic Editor

PLOS ONE

Journal Requirements:

Reviewers' comments:

Reviewer's Responses to Questions

**Comments to the Author**

1. If the authors have adequately addressed your comments raised in a previous round of review and you feel that this manuscript is now acceptable for publication, you may indicate that here to bypass the “Comments to the Author” section, enter your conflict of interest statement in the “Confidential to Editor” section, and submit your "Accept" recommendation.

Reviewer #2: All comments have been addressed

2. Is the manuscript technically sound, and do the data support the conclusions?

Reviewer #2: Yes

3. Has the statistical analysis been performed appropriately and rigorously? 

Reviewer #2: Yes

4. Have the authors made all data underlying the findings in their manuscript fully available?

Reviewer #2: Yes

5. Is the manuscript presented in an intelligible fashion and written in standard English?

Reviewer #2: Yes

6. Review Comments to the Author

Reviewer #2: Dear Author, I'm glad to see that you addressed all my previous comments.

I just want to point out that the glossary I was referring to was about network metrics (i.e., transitivity, density, assortativity, etc.) and not about the different bibliometric networks which are instead exhaustively explained within the text. Network metrics are indeed popular in network theory, but they may not be trivial for any audience. Since you mention these metrics within the text before introducing them properly, I suggested to include a glossary of network metrics to explain what they are and what they assess.

Moreover, I appreciated a lot the intregration of co-word and LDA analysis in your study. I would just suggest you to stress more the key differences in the clusters you identified through publication authorship and by the application of these alternative methods in the conclusion.

Once you fix these very minor aspects, the paper is ready to be accepted from my side.

7. PLOS authors have the option to publish the peer review history of their article (what does this mean?). If published, this will include your full peer review and any attached files.

Reviewer #2: No

---

## [Author Response · Author response to Decision Letter 2]

13 Dec 2023

Again, thank you for your comments on my manuscript. I have happily taken up your suggestions:

(1) The glossary now includes all network metrics used in the manuscript (betweenness, closeness, density, etc.).

(2) The conclusion now stresses the key differences between publication au- thorship and Latent Dirichlet Allocation (LDA) in terms of methodology and the clusters found by means of one or the other approach.

Moreover, the journal requirements asked to review the reference list. I can safely say that the references are complete and correct.

With these changes to the manuscript, I hope to have satisfied your critique!

---

## [Editor Report · Decision Letter 3]

27 Dec 2023

Publication authorship: A new approach to the bibliometric study of scientific work and beyond

PONE-D-22-17286R3

Dear Dr. Blaschke,

We’re pleased to inform you that your manuscript has been judged scientifically suitable for publication and will be formally accepted for publication once it meets all outstanding technical requirements.

Kind regards,

Roy Cerqueti, Ph.D.

Academic Editor

PLOS ONE
---

## [Editor Report · Acceptance letter]

7 Feb 2024

PONE-D-22-17286R3 

PLOS ONE

Dear Dr. Blaschke, 

I'm pleased to inform you that your manuscript has been deemed suitable for publication in PLOS ONE. Congratulations! Your manuscript is now being handed over to our production team.

Kind regards, 

on behalf of

Professor Roy Cerqueti 

Academic Editor

PLOS ONE